# Towards GAN Benchmarks Which Require Generalization

**Ishaan Gulrajani**
Google Brain
igul222@gmail.com

**Colin Raffel**
Google Brain
craffel@gmail.com

**Luke Metz**
Google Brain
lmetz@google.com

## Abstract

For many evaluation metrics commonly used as benchmarks for unconditional image generation, trivially memorizing the training set attains a better score than models which are considered state-of-the-art; we consider this problematic. We clarify a necessary condition for an evaluation metric not to behave this way: estimating the function must require a large sample from the model. In search of such a metric, we turn to neural network divergences (NNDs), which are defined in terms of a neural network trained to distinguish between distributions. The resulting benchmarks cannot be "won" by training set memorization, while still being perceptually correlated and computable only from samples. We survey past work on using NNDs for evaluation, implement an example black-box metric based on these ideas, and validate experimentally that it can measure a notion of generalization.

## 1 Introduction

In machine learning, it is often difficult to directly measure progress towards our goals (e.g. "classify images correctly in the real world", "generate valid translations of text"). To enable progress despite this difficulty, it is useful to define a standardized *benchmark task* which is easy to evaluate and serves as a proxy for some *final task*. This enables much stronger claims of improvement (albeit towards a somewhat artificial task) by reducing the risk of inadequate baselines or evaluation mistakes, with the hope that progress on the benchmark will yield discoveries and methods which are useful towards the final task. This approach requires that a benchmark task satisfy two properties:

1. It should define a straightforward and objective evaluation procedure, such that strong claims of improvement can be made. Any off-limits methods of obtaining a high score (e.g. abusing a test set) should be clearly defined.

2. Improved performance on the benchmark should require insights which are likely to be helpful towards the final task. The benchmark should, by construction, reflect at least some of the kinds of difficulty inherent in the final task.

Together these imply that a benchmark should at least be *nontrivial*: we should not know *a priori* how to obtain an arbitrarily high score, except perhaps by clearly-off-limits methods. Crucial to a useful benchmark is an *evaluation metric* which measures what we care about for the final task and which satisfies the requirements outlined above.

This paper deals with unconditional generation of natural images, which has been the goal of much recent work in generative modeling (e.g. Radford et al., 2015; Karras et al., 2017). Our ideas are general, but we hope to improve evaluation practice in models like Generative Adversarial Networks (GANs) (Goodfellow et al., 2014). Generative modeling has many possible final tasks (Theis et al., 2015). Of these, unconditional image generation is perhaps not very useful directly, but the insights and methods discovered in its pursuit have proven useful to other tasks like domain adaptation (Shrivastava et al., 2017), disentangled representation learning (Chen et al., 2016), and imitation learning (Ho & Ermon, 2016).

Some notion of *generalization* is crucial to why unconditional generation is difficult (and hence interesting). Otherwise, simply memorizing the training data exactly would yield perfect "generations"

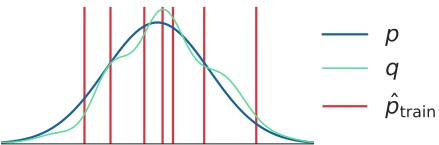

Figure 1: Common GAN benchmarks prefer training set memorization ($\hat{p}_{\text{train}}$, red) to a model ($q$, green) which imperfectly fits the true distribution ($p$, blue) but covers more of $p$'s support.

Table 1: The Inception Score and FID assign better scores to memorization of the training set, but a neural network divergence, $D_{\text{CNN}}$, prefers a GAN which generalizes beyond the training set.

| EVAL. METRIC | GAN | MEMORIZATION |
|---|---|---|
| INCEP. SCORE ↑ | 6.49 | **11.3** |
| FID (TRAIN) ↓ | 38.6 | **0.51** |
| FID (TEST) ↓ | 38.6 | **5.63** |
| $D_{\text{CNN}}$ (TRAIN) ↓ | 12.8 | **1.69E-4** |
| $D_{\text{CNN}}$ (TEST) ↓ | **12.9** | 14.7 |

and the task would be meaningless. One might argue that some work—particularly recent GAN research—merely aims to improve convergence properties of the learning algorithm, and so generalization isn't a big concern. However, generalization is an important part of why GANs themselves are interesting. A GAN with better convergence properties but no ability to generalize is arguably not a very interesting GAN; we believe our definition of the task, and our benchmarks, should reflect this.

Because our task is to generate samples, and often our models only permit sampling, recent work (e.g. Huang et al., 2016; Gulrajani et al., 2017; Tolstikhin et al., 2017) has adopted benchmarks based on evaluation metrics (Salimans et al., 2016; Heusel et al., 2017) which measure the perceptual quality and diversity of samples. However these particular metrics are, by construction, trivially "won" by a model which memorizes the training set. In other words, they mostly ignore any notion of generalization, which is central to why the task is difficult to begin with. This idea is illustrated in Figure 1. In this sense they give rise to "trivial" benchmark tasks which can lead to less convincing claims of improvement, and ultimately less progress towards useful methods. While "nontrivial" benchmark tasks based on downstream applications can be used, the goals of these tasks are at best indirectly related to, and at worst opposite from, sample generation (e.g. for semi-supervised learning (Dai et al., 2017)).

This paper considers evaluation metrics for generative models which give rise to nontrivial benchmarks, and which are aligned with the final task of generating novel, perceptually realistic and diverse data. We stress the difference between evaluating models and defining benchmarks. The former assumes that models have been chosen ahead of time, independently of the metric. The latter assumes that models will be developed with the metric in mind, and so seeks to avoid falsely high scores resulting from exploiting undesirable solutions to the benchmark (e.g. memorizing the training set). Our goal is to define benchmarks, and many of our decisions follow from this. Our contributions are as follows:

- We establish a framework for sample-based evaluation that permits a meaningful notion of generalization. We clarify a necessary condition for the ability to measure generalization: That the evaluation requires a large sample from the model.

- We investigate using *neural network divergences* (Arora et al., 2017) as evaluation metrics which have attractive properties for this application. We survey past work exploring the use of neural network divergences for evaluation.

- We study an example neural network divergence called "CNN divergence" ($D_{\text{CNN}}$) experimentally. We demonstrate that it can detect and penalize memorization and that it measures diversity relatively more than other evaluation functions.

## 2 BACKGROUND

Throughout the paper we cast evaluation metrics as statistical divergences specifying a notion of dissimilarity between distributions. The goal of generative modeling, then, is minimizing some divergence, and the choice of divergence reflects the properties of our final task.

Let $\mathcal{X}$ denote a set of possible observations (a common choice for image modeling is $\mathbb{R}^{32 \times 32 \times 3}$), and $\mathcal{P}(\mathcal{X})$ the set of probability distributions on $\mathcal{X}$. The goal of generative modeling is: given a data

distribution $p \in \mathcal{P}(\mathcal{X})$ (which we can only access through a finite sample) and a set of distributions $\mathcal{Q} \subseteq \mathcal{P}(\mathcal{X})$ (usually a parametric model family), find $q^* \in \mathcal{Q}$ which is closest to $p$ according to some definition of closeness. A notion of closeness between distributions is characterized by a statistical divergence, which is a function $D : \mathcal{P}(\mathcal{X}) \times \mathcal{P}(\mathcal{X}) \to \mathbb{R}_{\geq 0} \cup \{+\infty\}$ satisfying $D(p, q) = 0$ iff $p = q$.[1] Usually we cannot exactly evaluate $D(p, q)$, and so for training and evaluation purposes we must compute an estimate using finite samples from $p$ and $q$. We denote such an estimate of $D$ as $\hat{D}$. With some abuse of notation, we denote by $\hat{p}$ either a finite sample from $p$ or the empirical distribution corresponding to that sample, and likewise for $\hat{q}$ and $q$.

## 2.1 Evaluation Metrics for Models of Natural Images

Various evaluation criteria have been proposed for evaluating a generative model of natural images solely based on a finite sample from the model. Perhaps the most commonly applied metrics are the "Inception Score" (IS) (Salimans et al., 2016) and "Fréchet Inception Distance" (FID) (Heusel et al., 2017). These metrics, described in detail in Appendix B and Appendix C respectively, compute statistics on features produced by an Inception network (Szegedy et al., 2016) trained on ImageNet (Deng et al., 2009). They are motivated by the intuition that statistics computed on samples from the model $q$ should match those of real data. Apart from the IS and FID, test set log-likelihood is a popular metric for evaluation, but is not directly computable for models from which we can only draw samples. While Wu et al. (2016b) proposed a method of estimating log-likelihoods for "decoder-based" generative models, it requires additional assumptions to be made and was shown to be a poor estimate in some cases (Grover et al., 2017). Log-likelihood has also been shown to not correlate well with downstream needs such as sample quality (Theis et al., 2015).

## 3 Generalization in Sample-based Evaluation

In this section we establish a framework for sample-based evaluation that permits a meaningful notion of generalization. We state a property of divergences which are estimable from a finite sample (that they're somewhat insensitive to diversity), propose a baseline for non-trivial performance (the model should outperform training set memorization), explain that beating the baseline might only be achievable under certain kinds of divergences (the divergence should consider a large model sample), and finally explain that minimizing such a divergence requires paying attention to generalization.

## 3.1 Easily-Estimated Divergences Can't Measure Diversity

Because we can only sample from our models, and we have finite compute, we are restricted to divergences which can be estimated using only a finite sample from the model. We begin with an obvious but important note: any divergence estimated from a sample can be "fooled" by a model which memorizes a finite sample of not much greater size. Specifically:

**Remark 1.** *If $D(p, q)$ is a divergence which can always be estimated with error less than $\epsilon$ using only $m$ points sampled from $q$[2], $q_{\text{good}}$ is a model distribution, and $q_{\text{bad}}$ is an empirical version of $q_{\text{good}}$ with $m^2$ points, then $|D(p, q_{\text{good}}) - D(p, q_{\text{bad}})| < 2\epsilon$.*

In this case, we say that $D$ is (to some extent) insensitive to "diversity", which we define as any differences between a distribution and its empirical counterpart. The reasoning is that a sample of $m$ points (with replacement) from $q_{\text{bad}}$ is quite likely to also be a valid sample from $q_{\text{good}}$ (because it's unlikely that a point in $q_{\text{bad}}$ gets sampled twice), making it impossible to reliably tell whether the sample came from $q_{\text{good}}$ or $q_{\text{bad}}$. So, either the estimates must sometimes have substantial error or the difference between the true divergences must also be small.

This comes almost directly from Gretton et al. (2012)'s Example 1 and is similar in spirit to Arora et al. (2017)'s Corollary 3.2. This version requires that $D$ can be estimated with bounded error for a finite sample size, which is slightly too restrictive. Nonetheless it clarifies the intuition: *the sample size required to estimate a divergence upper-bounds the divergence's sensitivity to diversity.*

---

[1] In this work we permit $D(p, q) = 0$ for some $q \neq p$, since these cases aren't relevant if we assume our model $q$ is never good enough to exactly attain $D(p, q) = 0$.

[2] By this, we mean that there exists a function $\hat{D}$ satisfying $|\hat{D}(p, \hat{q}_m) - D(p, q)| < \epsilon$ for all $p, q$.

### 3.2 Models Should Outperform Training Set Memorization

From the previous section, we know that even trivially memorizing the training set (that is, choosing $q = \hat{p}_{\text{train}}$) is likely to minimize $D(p, q)$ (to an extent depending on $D$ and the size of $\hat{p}_{\text{train}}$) because $\hat{p}_{\text{train}}$ is a finite sample from $p$. Such memorization, however, is presumably not a useful solution in terms of our final task. Therefore we propose that in order for a model to be considered useful, it should at least outperform that trivial baseline. Specifically:

**Definition 1.** *A model $q$ outperforms training set memorization under a divergence $D$ if $D(p, q) < D(p, \hat{p}_{\text{train}})$.*

Cornish et al. (2018) propose essentially the same criterion. The next section explains how to choose $D$ such that this goal is achievable.

### 3.3 The Divergence Should Require a Large Sample (But Not Too Large)

Our goal is to find a model $q$ which outperforms training set memorization under $D$, but of course this depends on our choice of $D$. Intuitively, such a model makes $D(p, q)$ small by recovering some of the diversity which exists in the true distribution, but not the training set. But if $D$ isn't sensitive enough to diversity, very few models may exist which outperform $\hat{p}_{\text{train}}$ under $D$. Specifically, Remark 1 implies (by taking $q_{\text{good}} = p$ and $q_{\text{bad}} = \hat{p}_{\text{train}}$) that when $D$ can be estimated to small error using a sample of $m^{1/2}$ points ($m$ being the size of $\hat{p}_{\text{train}}$), $D(p, \hat{p}_{\text{train}})$ will be very close to zero. This suggests that finding a model satisfying $D(p, q) < D(p, \hat{p}_{\text{train}})$ might be very difficult when $D$ can be estimated with a small sample.

This motivates using divergences under which $D(p, \hat{p}_{\text{train}})$ is not a good solution. Such divergences require a large sample size (relative to the size of $\hat{p}_{\text{train}}$) to estimate, but fortunately this is often feasible. For example, the CIFAR-10 training set contains 50 thousand images, but we can sample 50 *million* images from a typical GAN generator within a reasonable computational budget.

Usually the underlying distribution we are dealing with is assumed to be exponentially large (say, on the order of $10^{50}$ distinct points), which dwarfs the largest sample we can possibly draw from a model. As such, no purely sample-based evaluation will be able to tell whether the model actually covers the full distribution. However, this also means that no purely sample-based final task will depend on such coverage. In this sense our work both suggests what generalization should mean in the context of sample-based applications and addresses how to evaluate it.

At the same time, we must be careful that our choice of $D$ does not require a larger sample to estimate than we can draw from $q$; otherwise our empirical estimate might be unreliable. Poor estimates are known to be hazardous in this setting: for example, the empirical Wasserstein distance, which converges exponentially slowly in the dimension (Sriperumbudur et al., 2012), systematically prefers models which generate blurry images (Huang et al., 2017; Cornish et al., 2018).

### 3.4 Learning Should Require Small Generalization Error

It's worth mentioning how we plan to find a model which minimizes $D(p, q)$ given that we don't usually have direct access to $p$. Making guarantees is difficult without reference to a specific $D$ or a specific learning algorithm, but in general we might hope to apply the usual machine learning technique (e.g. Vapnik, 1995): optimize a (surrogate) training loss $\hat{D}(\hat{p}_{\text{train}}, q)$, use a held-out set to estimate $D(p, q)$, bound the *generalization error* $D(p, q) - \hat{D}(\hat{p}_{\text{train}}, q)$ by constraining model capacity, and finally pick the model which best minimizes our estimate of $D(p, q)$.

This might seem obvious, but the situation is much less clear when $q = \hat{p}_{\text{train}}$ nearly minimizes $D(p, q)$ (that is, when $D$ is insensitive to diversity). In this case, the usual motivation behind trying to bound generalization error – namely, minimizing $D(p, q)$ given only access to $\hat{p}_{\text{train}}$ – doesn't apply. For example, in an evaluation metric like FID, if $q = \hat{p}_{\text{train}}$ yields a near-optimal score *with respect to the test set*, then it's not clear what we aim to achieve through generalization, even if we might be able to measure it sometimes. We show in Figure 1 that FID does indeed behave this way. In this sense, we can say that only divergences which require a large model sample permit a "meaningful" notion of generalization.

## 4 NEURAL NETWORK DIVERGENCES AS BENCHMARK METRICS

A promising approach for comparing two distributions using finite samples from each are *neural network divergences* (NNDs) (Arora et al., 2017; Liu et al., 2017; Huang et al., 2017), which are defined in terms of the loss of a neural network trained to distinguish between samples from the two distributions. Liu et al. (2017) define the family of divergences as follows:

$$D_{\text{NN}}(p, q) = \sup_{f_\theta \in \mathcal{F}} \mathbb{E}_{(x_p, x_q) \sim p \otimes q} \left[ \Delta(f_\theta(x_p), f_\theta(x_q)) \right] \tag{1}$$

for some function $\Delta : \mathbb{R}^d \times \mathbb{R}^d \to \mathbb{R}$ (which, loosely, looks like a classification loss) and parametric set of functions $\mathcal{F} = \{f_\theta : \mathcal{X} \to \mathbb{R}^d; \theta \in \Theta\}$ (e.g. a neural network architecture). In practice, utilizing an NND to evaluate a generative model amounts to training an independent "critic" network whose objective is to distinguish between real and generated samples. After sufficient training, the critic's loss can be used as a score reflecting the discriminability of real and generated data. To use an NND as an evaluation metric, we draw samples from our learned generative model $q$ and utilize a held-out test set $\hat{p}_{\text{test}}$ of samples from $p$ which was not used to train the generative model.

There exists some recent work on using NNDs for generative model evaluation (Danihelka et al., 2017b; Rosca et al., 2017; Bowman et al., 2015). Danihelka et al. (2017b) show empirically that NNDs can detect overfitting, although their setup leaves open questions which we address in our experiments. More recently, Im et al. (2018) provided a meta-evaluation of evaluation methods arriving from using critics trained with different GAN losses and compared them to existing metrics.

A closely-related and complementary body of work proposes model evaluation through two-sample testing, likewise treating evaluation as a problem of discriminating between distributions. Sutherland et al. (2016) evaluate GANs using a test based on the MMD (Gretton et al., 2012), and find the MMD with a generic kernel isn't very discriminative. Ramdas et al. (2015) show that with generic kernels, the MMD test power tends to decrease polynomially in the dimension. Li et al. (2017) recover these by combining the MMD with a learned discriminator (although not for evaluation). Bellemare et al. (2017) propose a similar method; Bińkowski et al. (2018) connect the two to each other and to NNDs, proving that all the estimators are biased. Bińkowski et al. (2018) also propose evaluation using a generic MMD in a pretrained feature space; this works at least as well as the FID, but it's not clear whether pretrained features are discriminative enough to detect overfitting. Lopez-Paz & Oquab (2016) explore evaluation by a test based on a binary classifier. Among other options, they consider using a neural net as that classifier, which amounts to using an NND for evaluation, although they report difficulties due to problems since addressed by Arjovsky et al. (2017).

**Outperforming Memorization** We are interested in NNDs largely because they let us realize the ideas in Section 3. In particular, they satisfy the criterion of subsection 3.3: estimating one involves training a neural network, a process which can consider about $10^7$ data points — a sample large enough for the divergence to be able to measure diversity beyond the size of most training sets, but not so large as to be computationally intractable. Thus we might hope to establish that our models generalize meaningfully and outperform training set memorization in the sense of Definition 1.

Concretely, NNDs measure diversity through the ability of the underlying neural network to overfit to data. If $q$ is an empirical distribution with a small number of points, the network can overfit to those points and more easily tell them apart from the points in $p$, causing the resulting divergence to be higher. Moreover, by changing the capacity of the neural network (e.g. by adding or removing layers), we can tune the sample size at which it begins to overfit (and the sample complexity of the divergence) as we desire.

**Perceptual Correlation** Compared to traditional statistical divergences, NNDs have an advantage in their ability to incorporate prior knowledge about the task (Huang et al., 2017). For example, using a convolutional network (CNN) for the critic results in a shift-invariant divergence appropriate for natural images. To test the correlation of CNN-based NNDs to human perception, Im et al. (2018) generated sets of 100 image samples from various models and compared the pairswise preferences given by various NNDs to those from humans. They found that the NND metrics they studied agreed with human judgement the vast majority of the time. IS and FID are also perceptually correlated (Salimans et al., 2016; Im et al., 2018), but they use pretrained ImageNet classifier features, which may make them less applicable to arbitrary datasets and tasks. In contrast, NNDs are applicable

to any data that can be fed into a neural net; for example, Bowman et al. (2015) used an NND to evaluate a generative model of text.

## 4.1 DRAWBACKS OF NNDs FOR EVALUATION

**Training Against the Metric**   Prior work has argued (Arjovsky et al., 2017; Im et al., 2018) that using an NND for GAN evaluation can be hazardous because an NND closely relates to a GAN's discriminator, and thus might "unfairly" prefer GANs trained with a similar discriminator architecture. More broadly, optimizing a loss which is too similar to a benchmark metric can cause issues if the benchmark fails to capture properties required for the "final task". For example, in NLP, directly optimizing the BLEU or ROUGE scores on the training set tend to score higher on the benchmark without necessarily improving human evaluation scores (Wu et al., 2016a; Paulus et al., 2017).

Im et al. (2018) "investigated whether the [neural network divergence] metric favors samples from the model trained using the same metric" by training GANs with different discriminator objectives and evaluating them with NNDs utilizing the same set of objectives. Despite the potential for bias, they found the metrics largely agreed with each other. This suggests that the objectives might have reasonably similar minimizers, and that in some sense we are not "overfitting" to the benchmark when the training and test objectives are the same. However, it may nevertheless be that NNDs prefer generative models trained with a similar objective (i.e. GANs); we observe this in Section 5.

In contrast, training a generative model of images directly against metrics like the IS has been shown to produce noise-like samples which achieve extremely high scores (Barratt & Sharma, 2018). While training directly against the IS is arguably "off-limits", there has been some evidence of techniques accidentally overfitting to the score. For example, Shu et al. (2017) suggest that the IS improvements achieved by AC-GAN (Odena et al., 2016) may be caused by AC-GAN models being biased towards samples near the classifier decision boundary; that the IS prefers this is arguably an undesirable artifact of the score. The existence of such a failure mode suggests that using the IS as a benchmark is ill-advised. In contrast, we are not aware of a similar failure mode for NNDs, but we believe further research into the robustness of these scores is warranted.

**Sensitivity to Implementation Details**   In order for an evaluation metric to be used for benchmarking, the metric itself needs to be used consistently across different studies. If implementation details can cause the scores produced by a metric to vary significantly, this conflates comparison of when different implementations are used to compare different methods. This has caused issues in benchmarking machine translation (Post, 2018) and music information retrieval (Raffel et al., 2014). This problem is particularly pronounced for NNDs because they require implementing a neural network architecture and training scheme, and the use of different software frameworks or even driver versions can cause results to vary (Henderson et al., 2018; Oliver et al., 2018). We emphasize that if NNDs are to be used as a benchmark, it is absolutely crucial that the same implementation of the metric is used across studies.

**Bias From a Small Test Set**   As explained in subsection 3.3, if an NND requires a larger sample to estimate than we have, we end up with a potentially-hazardous biased estimate. While we can draw an arbitrarily large model sample, the data distribution sample (that is, the test set) is finite and often fairly small. In concrete terms: if our critic network is big enough to detect an overfit or "collapsed" generator by overfitting to the generator's samples, then the critic can also overfit to the test set. Fortunately, since this bias comes from the test set and not the model distribution, we might hope that the resulting biased estimates still rank models in the same order as the unbiased ones would; we verify in the experiments that this appears to be the case for our models and NND. Still, we'd like stronger guarantees that our estimates are reliable, either by designing an NND without this bias or by a more careful analysis of the nature bias.

## 4.2 AN EXAMPLE NEURAL NETWORK DIVERGENCE METRIC

To empirically investigate the use of NNDs for evaluation, we developed a simple architecture and training procedure which we refer to as CNN divergence ($D_{\mathrm{CNN}}$) which we describe in detail in Appendix D. As a short summary, our model architecture is a convolutional network taking $32 \times 32 \times 3$ images as input and closely resembles the DCGAN discriminator (Radford et al., 2015).

Table 2: For different evaluation metrics, how many training set images does one need to memorize to score better than a well-trained GAN model? A larger $n$ means the metric is more sensitive to diversity.

| Eval. Metric | Input Size | $n$ to Win |
|---|---|---|
| Incep. score | 50,000 | 32 |
| FID | 50,000 | 1024 |
| Small CNN Div. | 25M | 32,768 |
| CNN Div. | 25M | $> 1M$ |

Table 3: Evaluation of different models on CIFAR-10 by test set CNN divergence. The WGAN-GP attains a lower value of the test divergence than memorization of the training set.

| Method | $D_{CNN}(\hat{p}_{test}, q)$ |
|---|---|
| PixelCNN++ | 16.17 |
| IAF VAE | 18.11 |
| WGAN-GP | 12.97 |
| Training set | 14.62 |

To provide useful signal when the distributions being compared are dissimilar, we use the critic objective from WGAN-GP (Gulrajani et al., 2017) which doesn't saturate when the distributions are easily discriminable by a neural network (i.e. have nonoverlapping support). We also utilize a carefully-tuned learning rate schedule and an exponential moving average of model parameters for evaluation, which we show produces a low-variance metric in Appendix E Our goal in developing the CNN divergence is not to propose a new standardized benchmark, but rather to develop a reasonable testbed for experimenting with adversarial divergences in the context of the current study. To facilitate future work on NNDs, we make an example implementation of the CNN divergence available.[3]

## 5 Experiments

Here we present experiments evaluating our CNN divergence's ability to assess generalization.

**Detecting and Outperforming Memorization** We are first interested in measuring the extent to which different evaluation metrics prefer memorization or generalization. To do so, we trained a WGAN-GP (Gulrajani et al., 2017) on the CIFAR-10 dataset and evaluated the resulting samples using IS, FID, and the CNN divergence using both $\hat{p}_{train}$ and $\hat{p}_{test}$ as our collections of samples from $p$. To compare the resulting scores to training set memorization, we also evaluated each metric on $\hat{p}_{train}$ itself. The results are shown in Table 1. We find that both the IS and the FID assign better scores to simply memorizing the training set, but that the CNN divergence assigns a worse (higher) score to training set memorization than the GAN. Further, the score $D_{CNN}(\hat{p}_{train}, \hat{p}_{train})$ is much smaller than $D_{CNN}(\hat{p}_{test}, \hat{p}_{train})$, which suggests the ability to detect overfitting. The results of this experiment are summarized schematically in Figure 1: The IS and FID prefer memorizing a small sample, whereas the CNN divergence prefers a model which imperfectly fits the underlying distribution but covers more of its support.

**Comparing Evaluation Metrics' Ability to Measure Diversity** For several evaluation functions, we test how many sample points from the true distribution are needed to attain a better score than a well-trained GAN model. We consider IS, Fréchet Inception Distance, our CNN divergence, and a modified version of the CNN divergence with a smaller critic network with $1/4$ as many channels at each layer. We begin by training a WGAN-GP (Gulrajani et al., 2017) on the $32 \times 32$ ImageNet dataset (Oord et al., 2016); we denote the model distribution by $q$. We expect this GAN to generate images which are less realistic than the training set, but reasonably diverse (at least on the order of $10^5$ perceptually unique images, based on Arora & Zhang (2017)). We estimate each evaluation metric $D(p, q)$ using a test set $\hat{p}_{test}$ and a sample from $q$. We similarly estimate $\mathbb{E}_{\hat{p}_n}[D(p, \hat{p}_n)]$, where $\hat{p}_n$ is a random $n$-point subset of $\hat{p}_{train}$, for many values of $n$. In Table 2 we report the smallest $n$ where memorizing $n$ training set images scores better than the GAN. Higher $n$ means that $D$ assigns relatively more importance to diversity.

Consistent with our expectation, we find that both CNN divergences require memorizing many more images to attain an equivalent score to a GAN, so Definition 1 is more likely to be achievable under these. Moreover, decreasing the capacity of the underlying critic network also decreases this

---

[3] https://github.com/google-research/google-research/tree/master/towards_gan_benchmarks

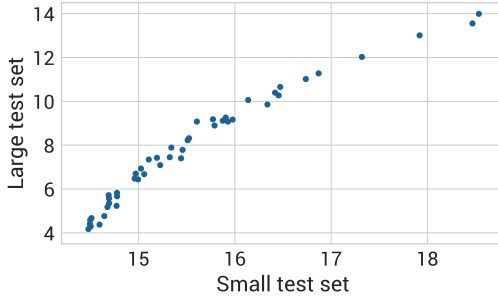 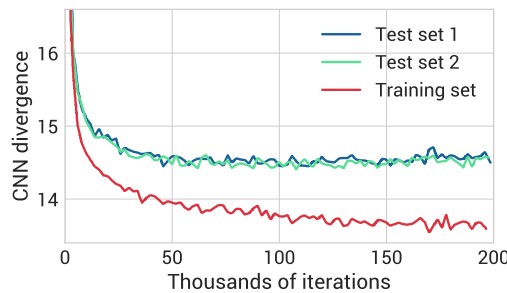

Figure 2: CNN divergence on a small test set is biased, but correlated, with a large test set.

Figure 3: CNN divergence reveals overfitting in a large GAN.

sensitivity to diversity. In particular, the standard CNN divergence penalizes memorization to a greater extent than the small-critic version.

**The Effect of Bias From a Small Test Set** Our evaluation function requires a very large test set to accurately estimate, and using a normally-sized test set yields a biased estimate. However, we might hope that because the bias comes from the test set and not the model, the small-test-set estimates might still rank different models in the same order as large-test-set estimates would. To check this, we split $32 \times 32$ ImageNet into training ($n$=50,000), "small test" ($n$=10,000), and "large test" ($n$=1,290,000) sets. We train 64 GAN models with randomly-chosen hyperparameters on the training set and evaluate their CNN divergence with respect to both test sets. We plot the results in Figure 2. We observe that the two values are strongly correlated over this set of models (up to the level of noise introduced by the optimization process), suggesting that it might be safe to use a small test set in this setting.

**Divergence Values Throughout Model Training** We train a somewhat large (18M parameters) GAN on a 50,000-image subset of $32\times32$ ImageNet. Every 2000 iterations, we evaluate three CNN divergences: first, with respect to a held-out test set of 10,000 images, second, another independent test set of the same size (to verify that the variance with respect to the choice of test set images is negligible), and last, a 10,000-image subset of the training set (we use a subset to eliminate bias from the dataset size). Each of the 300 resulting CNN divergence evaluations was run completely from scratch. We plot the results in Figure 3.

We first note that the values of all the divergences decrease monotonically over the course of training. If we assume that a good model gets steadily better at the final task over its training and ultimately converges, then the value of a good evaluation metric should reflect this pattern, as ours does. We further observe a substantial gap between the training and test set divergences. Consistent with subsection 3.4, this suggests that to score well, we need to pay some attention to generalization.

The convergence result is closely related to the one in Arjovsky et al. (2017), and the generalization result to the one in Danihelka et al. (2017b). A major difference is that compared to the "critic" networks used in those works, our CNN divergence is a black-box evaluation metric computed from scratch at each point in training. This rules out the possibility that the observed behavior comes from an artifact in the critic training process rather than an underlying property of the model distribution.

Further, the overfitting result in Danihelka et al. (2017b) comes from "misusing the independent critic" by evaluating it on empirical distributions it wasn't trained on: their estimation procedure is different for the training, validation, and test sets. This complicates reasoning about the bias of the resulting estimators (see subsection 4.1). In contrast, we simply estimate the same divergence with respect to three empirical distributions.

**Evaluating Models Against CNN Divergence** To test whether the CNN divergence prefers models trained on a similar objective, we use it to evaluate several different types of generative models. We train 3 models: a PixelCNN++ (Salimans et al., 2017), a ResNet VAE with Inverse Autoregressive Flow (IAF) (Kingma et al., 2016), and a DCGAN (Radford et al., 2015) trained with the WGAN-GP objective (Gulrajani et al., 2017). Training details are given in the appendix. As a baseline, we also

report the score attained by memorizing the training set. We list results in Table 3. Notably, the GAN outperforms both of the other models, which may simply be because its objective is much more closely related to the CNN divergence than maximum likelihood (as used by PixelCNN++ and the VAE); we discuss this effect in subsection 4.1. In addition, the GAN achieves a better score than memorizing the training set. This result satisfies Definition 1 and allows us to say that the GAN has achieved a nontrivial benchmark score through generalization.

## 6 CONCLUSION

We believe our experiments show that the NNDs are a promising direction for evaluating generative models. They are not trivially solved by memorizing the training set, which satisfies our argument (Section 3) that measuring generalization ability is linked to whether the metric requires a large collection of samples. They also appear to prefer diversity relatively more than the IS and FID metrics. We note that NNDs are almost certainly not the only evaluation metric under which models can meaningfully generalize, and encourage work on alternative metrics.

Ultimately, models should be evaluated according to their intended final task (Theis et al., 2015). In our work we assume that our final task is not usefully solved by memorizing the training set, but for many tasks such memorization is a completely valid solution. If so, the evaluation should reflect this: we do not need our model to be closer to the true distribution than the training set, Definition 1 does not apply, and we might be free to consider evaluations which look at only a small sample from the model.

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

## A  THE IMPORTANCE OF TRADEOFFS IN EVALUATION METRICS

Many evaluation metrics proposed for generative modeling are asymptotically consistent: that is, in the limit of infinite data and model capacity, the model which exactly recovers the data distribution will be the unique minimizer of the metric. The data log-likelihood and the MMD (given a suitable kernel) are examples of asymptotically consistent evaluation metrics. Given that in the limit, all such metrics agree with each other (and therefore optimizing them will produce the same result), we might wonder whether the precise choice of metric actually matters: maybe it's sufficient to arbitrarily pick a "generic" metric like the MMD and minimize it.

However, none of this applies when our model is misspecified (that is, our model isn't capable of exactly recovering the data distribution). In that case, the learning algorithm must choose between capturing different properties of the data distribution. Here, the tradeoffs made by different metrics – which properties of the distributions they assign relatively more and less importance to – can have a strong effect on the result.

Theis et al. (2015) provide an excellent overview of this problem and its implications for generative model evaluation. They demonstrate on toy data that in the regime of model misspecification, optimizing three different asymptotically consistent metrics (the log-likelihood, the MMD, and the Jensen-Shannon divergence) yields three different results. Further, considering the task of image generation, they show that log-likelihood and sample quality can be almost completely independent when the model is even very slightly suboptimal. Their work cautions against the use of "generic" metrics and concludes that generative models should always be evaluated using metrics whose tradeoffs are similar to the intended downstream task.

The examples in Theis et al. (2015) are mostly hypothetical, so we review a few instances of this problem in recent state-of-the-art generative models of images:

- Training two Real NVP models on CelebA using a log-likelihood objective and an NND results in qualitatively very different models. Each model scores better than the other in terms of the metric it was trained to minimize, and worse in terms of the other metric (Danihelka et al., 2017a; Grover et al., 2017).
- Two PixelCNN models with different architectures trained on CIFAR-10 can attain similar log-likelihoods very close to the state of the art, even while one generates much less realistic samples than the other (Salimans et al., 2017).
- By construction, the model architecture typically used in GAN generators attains the worst possible log-likelihood of negative infinity, even when the generator is well-trained, produces realistic samples, and is a good minimizer of an NND (Arjovsky & Bottou, 2017).

In this context, we take the view of Huang et al. (2017), who argue that NNDs are particularly good choices for evaluation metrics because we can control the tradeoffs they make by changing the discriminator architecture. For example, by using CNNs as discriminators, we can construct metrics which assign greater importance to perceptually relevant properties of the distribution of natural images.

## B  INCEPTION SCORE

The Inception Score $\text{IS}(q)$ (Salimans et al., 2016) is defined as

$$\text{IS}(q) = \mathbb{E}_{x \sim q}[D_{\text{KL}}(p(y|x)\|p(y))] \tag{2}$$

where $p(y|x)$ is output of an Inception network (Szegedy et al., 2016) trained on ImageNet (Deng et al., 2009), producing a distribution over labels $y$ given an image $x$ and $p(y)$ is the same but marginalized over generated images. While ad-hoc, it has been shown that this method correlates somewhat with perceived image quality (Salimans et al., 2016). To cast IS as a divergence, we will consider the absolute difference in scores between the data and the model:

$$D_{\text{IS}}(p, q) = |\text{IS}(p) - \text{IS}(q)| \tag{3}$$

The inception score is typically computed on 10 different samples of size 5,000, and the scores for each of the 10 samples are averaged to give a final score.

## C  FRÉCHET INCEPTION DISTANCE

The Inception Score only very indirectly incorporates the statistics of the real data. To mitigate this, Heusel et al. (2017) proposed the Fréchet Inception Distance (FID), which is defined as

$$D_{\text{FID}}(p, q) = \|\mu_p - \mu_q\|^2 + \text{Tr}\left(\Sigma_p + \Sigma_q - 2(\Sigma_p \Sigma_q)^{1/2}\right)$$

where $\mu_p, \Sigma_p$ and $\mu_q, \Sigma_q$ are the mean and covariance of feature maps from the penultimate layer of an Inception network for real and generated data respectively. This score was shown to be appropriately sensitive to various image degradations and better correlated with image quality compared to the Inception Score (Heusel et al., 2017). FID is typically computed using statistics from 50,000 real and generated images.

## D  CNN DIVERGENCE ARCHITECTURE AND TRAINING DETAILS

For the CNN divergence, we use a typical convolutional network architecture which consists of three $5 \times 5$ convolutional layers with $64$, $128$, and $256$ channels respectively. These layers are followed by a single fully-collected layer which produces a single scalar output. All convolutional layers utilize a stride of $2$ and are each followed by a Swish nonlinearity (Ramachandran et al., 2017). Parameters are all initialized using "He"-style initialization (He et al., 2015). We do not use any form of normalization, batch or otherwise.

The model is trained using the critic's loss from the WGAN-GP objective (Gulrajani et al., 2017). We train for 100,000 iterations using minibatches of size 256 with a learning rate of $2 \times 10^{-4}$. Our final loss value is computed after training, using an exponential moving average of model weights over training with a coefficient of $0.999$.

## E  VARIANCE OF CNN DIVERGENCE BETWEEN RUNS

To verify that the CNN divergence returns approximately the same value each time it is run with the same model, we train a single GAN on CIFAR-10 and then estimate the divergence $D_{\text{CNN}}(\hat{p}_{\text{train}}, q)$ $n = 50$ times. To limit noise in the estimation process, we initialize weights carefully, train using learning rate decay, evaluate using a moving average of network weights. The resulting values have a mean of 5.51 and standard deviation of 0.03, which is approximately 0.5% of the mean. We conclude that our divergence is reliable across runs. Note that we are evaluating a single pretrained model, and different training runs of the same model might yield different scores. However, we view this as arguably a fault of the model's training process moreso than the of evaluation metric.

## F  EXPERIMENTAL DETAILS

### F.1  COMPARING EVALUATION METRICS' ABILITY TO MEASURE DIVERSITY

For IS and FID, we use sample sizes 5K and 10K common in past work (the estimates don't change much beyond this size) and for CNN divergence, we use the largest sample size we can (>1M).

### F.2  EVALUATING MODELS AGAINST CNN DIVERGENCE

For the VAE and the PixelCNN++, we train 10 models using the hyperparameters and code provided by the authors and evaluate the divergence 10 times on each. For the GAN, we train 64 models, searching randomly over hyperparameters, and evaluate the divergence once per model. In all cases, we report the best score overall.

