# OpenReview forum: "Towards GAN Benchmarks Which Require Generalization"
_ICLR.cc/2019/Conference_

### Official Review · AnonReviewer1 · 2018-11-01
**Not a thorough paper**

**Rating:** 3
**Confidence:** 4

**Review:**

The paper aims to come up with a criterion for evaluating the quality of samples produced by a Generative Adversarial Network. The main goal is that the criterion should not reward trivial sample generation algorithms such as the one which generates samples uniformly at random from the samples in the training set. I personally feel that if sample generation is the only goal, then this trivial algorithm is perfectly fine because, statistically, the empirical distribution is in many, though not all, ways, a good estimator of the underlying true probability measure (this is the idea that is used in the statistical technique of Bootstrap for example). However the underlying goal in unsupervised learning problems where GANs are used is hardly sample generation. The GANs also output a whole function in the form of a generative network which converts random samples into samples from the underlying generating distribution. This generative network is arguably more important and more useful than just the samples that it generates. An evaluation scheme for GANs should focus on the generative network directly rather than on a set of its generating samples.

Even if one were to regard the premise of the paper as valuable, the paper still does a poor job meeting its objective. A measure D_CNN is proposed as a benchmark. It must be remarked that D_CNN is not even properly defined (for example, there is a function \Delta in its definition but it is never explained what this function is). D_CNN is a variant of the existing notion of Neural Network Divergences. Only a numerical study (with no theory) is done to illustrate the utility of D_CNN for evaluating samples generated by GANs. The entire paper is very anecdotal with very little rigorous theory.

---

> ### Author Response · Authors · 2018-11-23
> **Response to Reviewer 1**
>
> Thank you very much for the review. We'd like to respond as follows:
>
> > I personally feel that if sample generation is the only goal, then this trivial algorithm is perfectly fine because, statistically, the empirical distribution is in many, though not all, ways, a good estimator of the underlying true probability measure (this is the idea that is used in the statistical technique of Bootstrap for example).
>
> We absolutely agree! We write in the final paragraph "In our work we assume that our final task is not usefully solved by memorizing the training set, but for many tasks such memorization is a completely valid solution. If so, the evaluation should reflect this..."
>
> > However the underlying goal in unsupervised learning problems where GANs are used is hardly sample generation. The GANs also output a whole function in the form of a generative network which converts random samples into samples from the underlying generating distribution. This generative network is arguably more important and more useful than just the samples that it generates. An evaluation scheme for GANs should focus on the generative network directly rather than on a set of its generating samples.
>
> We agree that learning a generative network with a specific structure is a very important task in unsupervised learning. The argument that GAN research should be steered away from sample generation is certainly interesting. However without taking an opinion on that argument, we observe that a significant number of strong papers have been oriented at the final task of unconditional sample generation (e.g. https://arxiv.org/abs/1710.10196, ICLR 2017 oral). Since presumably this trend will continue, we believe that it’s valuable to work towards proper benchmarks for this task. And developing proper benchmarks requires a definition of the task which is nontrivial, i.e. for which training set memorization isn’t a perfect solution.
>
>
> > A measure D_CNN is proposed as a benchmark. It must be remarked that D_CNN is not even properly defined (for example, there is a function \Delta in its definition but it is never explained what this function is).
>
> We give a detailed specification of D_CNN in Appendix D, and we’re releasing code along with this paper which will serve as a canonical reference. However, we think of our D_CNN as an example instantiation of the idea of NNDs -- as such, we don’t think the specifics are relevant to most of our experiments or conclusions.
>
> > D_CNN is a variant of the existing notion of Neural Network Divergences. Only a numerical study (with no theory) is done to illustrate the utility of D_CNN for evaluating samples generated by GANs. The entire paper is very anecdotal with very little rigorous theory.
>
> We see sections 2-4 of our paper as a unification and expansion of existing theory from the particular point of view of whether an evaluation metric requires a large sample to be evaluated and whether neural network divergences satisfy this property. We believe this is a useful contribution which stands apart from the empirical results we present in Section 5.

---

### Official Review · AnonReviewer3 · 2018-11-02
**Well written overview of GAN benchmarks**

**Rating:** 7
**Confidence:** 4

**Review:**

Summary:
The paper looks at the problem of benchmarking models that unconditionally generate images. In particular they focus on GAN models and discuss the Inception Score (IS) and Fréchet Inception Distance (FID) metrics. The authors argue that a good benchmark should not have a trivial solution (e.g. memorising the dataset) and find that a necessary condition for such a metric is a large number of samples. They also find that for IS and FID , a GAN is outperformed by a model that memorises the dataset, while a method based on neural network divergences (NND) does not show the same behaviour. NND works by training a discriminative model to discriminate between samples of the generative model and samples from a held out test set. The poorer the discriminative model performs, the better the generative model is.

The authors show a range of results using a CNN based divergence: on PixelCNN++, GANs, overfitted GANs, WGAN-GP and conclude that it’s a better metric than IS/FID at the expense of requiring much more computation to evaluate.  They also perform a test with limited compute and show that the results correlate well with a bigger dataset, but show some bias.

Review:
The paper is well written, with a clear description of the properties a good benchmark should have, an analysis of the current solutions and their shortcomings and an extensive experimental evaluation of the CNN divergence metric. The authors also compared with non GAN methods and experimented with small datasets, both are not necessarily within scope but a welcome addition. The authors also open source their code.

In the section “Outperforming Memorization”, the authors mention a way to tune capacity of the “critic” network and influence its ability to overfit on the sample. This means that if someone wants to compare the generalisation and diversity of samples between GANs, they would need to train the exact same critic CNN to be able to make a comparison. However the authors do not provide any principled way to determine the right size of the "critic" network. In general, given evaluating the metric requires training a network from scratch, it will be very difficult to make this consistent. This makes the proposed benchmark more impractical to use than its alternatives.

In the section “training against the metric”, the authors mention that a main criticism is the fact that a GAN directly optimises for the NND loss. In table 3 we indeed see that this is the case, however the authors argue that perhaps the GAN is simply the better model. I am worried by the fact that both PixelCNN++ and IAF-VAE perform worse than the training set on this benchmark. It seems like this particular benchmark would then work well specifically for GANs, but would (still) not allow us to compare with models trained using maximum likelihood.

In conclusion, I think the paper is well written and the authors clearly make progress towards a dependable benchmark for GANs. The paper does not introduce any new method, but instead has a thorough analysis and discussion of current methods which is worthwhile by itself.

Nits:
Page 7, second paragraph, fifth line, spurious “q”

########
Revision

I would like to thank the authors for a thoughtful revision and response. I have updated my score to a 7 and think this paper is a worthy contribution to ICLR. The new drawback section is well written and informative.

---

> ### Author Response · Authors · 2018-11-23
> **Response to Reviewer 3**
>
> Thanks for taking the time to write this review. We'd like to respond to your points as follows:
>
> > (...) experimented with small datasets, both are not necessarily within scope but a welcome addition
>
> We'd like to clarify why we consider the small-test-set experiment to be a crucial contribution. We've updated the paper (sections 3.3 and 4.1) to explain that a small test set might be hazardous specifically for NNDs, which are designed to require a large sample to estimate. Without evidence that the small-sample estimates correlate very well with the large-sample estimates, we wouldn't effectively be able to use NNDs for evaluation except in settings where our test set is much larger than our training set.
>
> > if someone wants to compare the generalisation and diversity of samples between GANs, they would need to train the exact same critic CNN to be able to make a comparison. (...) In general, given evaluating the metric requires training a network from scratch, it will be very difficult to make this consistent.
>
> You’re absolutely right that it’s very difficult to reproduce network training identically across implementations and hardware. We’ve added a discussion of this problem in a section titled “Drawbacks of NNDs for Evaluation”. In short, NND-based evaluation will likely require standardized open-source hardware-independent implementations. In general, we don’t claim to have complete solutions for these problems - instead, we present a framework and a path forward for evaluating generative models based on samples alone. However, we do note that for our specific metric, CNN Divergence, the variance across multiple training runs of the critic network is quite small, as outlined in Appendix E.
>
> > However the authors do not provide any principled way to determine the right size of the "critic" network.
>
> Ultimately, the best critic size will depend on the downstream application of the generative model. Since this downstream task is usually not well-defined theoretically, determining the “right” critic size by theory is a very difficult task and it’s perhaps best left as an empirical choice. More generally, we avoid attempting to prescribe hyperparameters or define a specific evaluation procedure in this work.
>
> > In table 3 we indeed see that this is the case, however the authors argue that perhaps the GAN is simply the better model.
>
> This is a very important point; thanks for raising it. To clarify, we don't mean to suggest that the GAN is the "best" model in any aboslute sense. Instead, it simply is the model that performs best in terms of the CNN divergence. We believe the CNN divergence is more sensitive to certain properties of the learned distribution than, for example, log likelihood. Whether this means the GAN is better or worse will depend on the intended use of the generative model. We discuss this in a few places in the paper:
>
> In Section 4.1, “Training Against The Metric”, we argue that the NND’s tendency to “unfairly” favor models trained against it appears to be mild compared to metrics like the Inception Score, which very greatly favors models trained against it, even though those models produce samples which resemble pure noise.
> In Appendix A (newly added), we summarize and highlight new evidence for past arguments against any universal notion of a “best” metric or model: in short, different metrics always tend to prefer different models.
>
> We note that some studies (e.g. https://arxiv.org/abs/1705.05263,  https://arxiv.org/abs/1705.08868) have considered the performance of models trained against an NND in terms of log-likelihood by using a flow-based (invertible) generator and found that GAN training performs very poorly in terms of likelihood. This is a similar point to the one we make here - a model trained against one class of objective (e.g. via maximum likelihood) might not be expected to perform well against another class of objectives.
>
> > I am worried by the fact that both PixelCNN++ and IAF-VAE perform worse than the training set on this benchmark.
>
> Any useful metric will exhibit some trade-off between, for example, sample quality and diversity. In terms of why PixelCNN++ and IAF-VAE perform worse than the training set under CNN divergence, CNN divergence likely "prefers" sample quality to diversity to the extent that it prefers a small, perfectly realistic sample (i.e. the training set). We note that while the PixelCNN++ and IAF-VAE are certainly effective generative models, samples from those models are clearly distinguishable from the training set. We’ve updated the paper with a detailed discussion of this topic in Appendix A.
>
> > Nits
>
> Thanks for catching this! We’ve fixed it in an update.

---

### Official Review · AnonReviewer2 · 2018-11-04
**Good beginning. Algorithm is not that interesting**

**Rating:** 6
**Confidence:** 4

**Review:**

This paper is quite interesting as it tries to find a new metric for evaluating GANs. IS is a terrible metric, as memorization would achieve high score and test log-likelihood cannot be evaluated. I like the long discussion at the beginning of the paper about what a metric for evaluating implicit generative models would need to be a valid and useful metric. This problem is of great importance for GANs as proving that GANs solve the density estimation problem would be extremely hard and even more so, making sure we are close to a good solution with any finite sample even more so (I am talking to non-trivial examples in high dimensions). It is clear that in order to make GANs, in particular, or implicit models, in general, useful, we need to find metrics that would allow us to achieve progress. This paper is a direction in what it needed. In this sense I think the paper can be a good starting point for the discussion that we are not having right now, because we are too focused on making sure they converge, but not how they can be useful.

On the down side, I think the proposed DNN metric is not exactly useful. It would be a subset of the metric that an MMD would give and it would focus only in some properties of the images but not on the whole distribution. So, if this metric does not capture the relevant aspects of the problem the GAN is trying to imitate, it will fail to provide that metric that we are looking for.

I would see this paper as a great workshop paper, in the sense of old-fashion NIPS workshops in which new ideas were tested and discussed. But it clearly would like the polished papers that we see in conferences these days. Bernhard Schoelkopf told me once, after receiving the ICML reviews, “People now focus more on reasons to reject a paper than in reason for accepting a paper.” (note that I am quoting from memory, the bad use of English in mine not his). There are many reasons to reject this paper, but also some reason to accept the paper.

---

> ### Author Response · Authors · 2018-11-23
> **Response to Reviewer 2**
>
> Thank you for the thoughtful review! We'd like to respond to one point in particular:
>
> > On the down side, I think the proposed DNN metric is not exactly useful. It would be a subset of the metric that an MMD would give and it would focus only in some properties of the images but not on the whole distribution. So, if this metric does not capture the relevant aspects of the problem the GAN is trying to imitate, it will fail to provide that metric that we are looking for.
>
> We agree completely that NNDs have inductive biases which cause them to ignore certain properties of the distribution: for example, our “CNN divergence” is likely to ignore small spatial shifts in its inputs. However, we actually see this as an advantage: NNDs let us design metrics which are sensitive only to the properties that are important for the final task, and invariant to the rest. We think this point is best made by Theis et al. (https://arxiv.org/abs/1511.01844), who argue that evaluation metrics should reflect the downstream task, and Huang et al. (https://arxiv.org/abs/1708.02511), who argue theoretically and empirically that NNDs in particular are good losses for generative modeling *because* of their inductive biases. We addressed this briefly in our section titled “Perceptual Correlation”, but we think it definitely deserves a longer discussion -- so we’ve updated the paper with a separate section (Appendix A) which clarifies this point in detail with examples and references.
>
> Concerning the MMD in particular, we review past work on its use for model evaluation in section 4. We’ve updated that paragraph to add an important point: the MMD with a generic kernel tends not to be very discriminative in high dimensions. Reddi et al. (https://arxiv.org/abs/1406.2083) show that the power of a two-sample test based on the MMD decreases polynomially in high dimensions, for many types of distributions. NNDs, on the other hand, leverage the inductive biases of neural networks in order to produce a discriminative metric even in high dimensions.

---

### Author Response · Authors · 2018-11-23
**Thank you for the reviews! Paper updated.**

We’d like to thank all the reviewers for your thoughtful comments. We’ve made the following significant updates to our paper based on your feedback:

- Clarified throughout that our goal is to present a promising approach and motivate future work, rather than directly to propose a benchmark. To that end, added section 4.1, "Drawbacks of NNDs for Evaluation".
- Added a detailed discussion of the need for evaluation metrics tailored to a specific task in Appendix A, "The Importance of Tradeoffs in Evaluation Metrics".
- Clarified the situation with bias from a small test set in sections 3.3 and 4.1

Additionally, we've responded to your comments individually below.

---

### Meta-Review · Area_Chair1 · 2018-12-14
**Tackles an important problem with arguable success**

**Confidence:** 4
**Recommendation:** Accept (Poster)

**Metareview:**

The paper argues for a GAN evaluation metric that needs sufficiently large number of generated samples to evaluate. Authors propose a metric based on existing set of divergences computed with neural net representations. R2 and R3 appreciate the motivation behind the proposed method and the discussion in the paper to that end. The proposed NND based metric has some limitations as pointed out by R2/R3 and also acknowledged by the authors -- being biased towards GANs learned with the same NND metric; challenge in choosing the capacity of the metric neural network; being computationally expensive, etc. However, these points are discussed well in the paper, and R2 and R3 are in favor of accepting the paper (with R3 bumping their score up after the author response).
R1's main concern is the lack of rigorous theoretical analysis of the proposed metric, which the AC agrees with, but is willing to overlook, given that it is nontrivial and most existing evaluation metrics in the literature also lack this.
Overall, this is a borderline paper but falling on the accept side according to the AC.